# Year-round water management for desert bighorn sheep corresponds with visits by predators not bighorn sheep

Grant M. Harris[1]*, David R. Stewart[1], David Brown[2], Lacrecia Johnson[3], Jim Sanderson[4], Aaron Alvidrez[5], Tom Waddell[6], Ron Thompson[7]

1 United States Fish and Wildlife Service, Albuquerque, NM, United States of America, 2 Natural History Collections, Arizona State University, Tempe, AZ, United States of America, 3 United States Fish and Wildlife Service, Tucson, AZ, United States of America, 4 Global Wildlife Conservation, Austin, TX, United States of America, 5 Luke Air Force Base, Glendale, AZ, United States of America, 6 Tucson, AZ, United States of America, 7 Primero Conservation.org, Pinetop, AZ, United States of America

* grant_harris@fws.gov

## Abstract

Managing water (e.g., catchments) to increase the abundance and distribution of game is popular in arid regions, especially throughout the southwest United States, where biologists often manage water year-round for desert bighorn sheep (*Ovis canadensis nelsoni*). Bighorn may visit water when predators (e.g., mountain lions [*Puma concolor*], coyotes [*Canis latrans*]) do not, suggesting that differences in species ecology or their surface water requirements influence visit timing. Alternatively, visits by desert bighorn sheep and predators may align. The former outcome identifies opportunities to improve water management by providing water when desert bighorn sheep visit most, which hypothetically may reduce predator presence, range expansion and predation, thereby supporting objectives to increase sheep abundances. Since advancing water management hinges on understanding the patterns of species visits, we identified when these three species and mule deer (*Odocoileus hemionus*) visited managed waters in three North American deserts (Chihuahuan, Sonoran, Mojave). We unraveled the ecological basis describing why visits occurred by associating species visits with four weather variables using multi-site, multi-species models within a Bayesian hierarchical framework (3.4 million images; 105 locations; 7/2009-12/2016). Desert bighorn sheep concentrated visits to water within 4–5 contiguous months. Mountain lions visited water essentially year-round within all deserts. Higher maximum temperature influenced visits to water, especially for desert bighorn sheep. Less long-term precipitation (prior 6-week total) raised visits for all species, and influenced mountain lion visits 3–20 times more than mule deer and 3–37 times more than sheep visits. Visits to water by prey were inconsistent predictors of visits to water by mountain lions. Our results suggest improvements to water management by aligning water provision with the patterns and ecological explanations of desert bighorn sheep visits. We exemplify a scientific approach to water management for enhancing stewardship of desert mammals, be it the southwest United States or arid regions elsewhere.

**Data Availability Statement:** All relevant data are within the manuscript and its Supporting Information files.

**Funding:** The author(s) received no specific funding for this work.

**Competing interests:** The authors have declared that no competing interests exist.

## Introduction

Supplying water (e.g., catchments, drinkers) to increase the abundance of game species and other wildlife forms a popular management approach in arid regions [1–3]. Specifically, in the southwest United States, game managers provide water year-round to raise population sizes and expand the distributions of desert bighorn sheep (*Ovis canadensis nelsoni* [4, 5]). Other sympatric species like mountain lions (*Puma concolor*) and coyotes (*Canis latrans*) which prey upon desert bighorn sheep visit these managed waters too [6]. Predators, however, may visit water when desert bighorn sheep do not. This outcome would suggest that managed waters may not be serving as attractants to prey that are subsequently being hunted by predators. Instead, ecological differences between predators and prey or their surface water requirements influence the timing of their visits to water. Such an outcome would identify opportunities to improve water management by providing water only when the target species (i.e., desert bighorn sheep) visit water most. Hypothetically, water closures during remaining periods may reduce the abundances and range expansions of non-target species (like predators) within desert bighorn sheep ranges. If so, such a strategy would support the management objective of increasing desert bighorn sheep abundances by reducing predation, while lowering the impetus for predator control, thereby lessening costs and political controversy. Alternatively, the timing of visits to water by predators and desert bighorn sheep may align. This outcome could indicate that prey are attracting predators to water, or that these species have ecological similarities and responses to surface water. Such an outcome would limit the potential to target the timing of water management to desert bighorn sheep. A first step towards answering these questions and informing the related hypotheses requires identifying when these species visit water and unraveling the ecological basis describing why the visits occur.

In the United States, providing water for desert ungulates like desert bighorn sheep occurs where free-standing water does not exist, becomes scarce during dry periods, or where anthropogenic actions eliminated animals' access [2, 3, 5, 7–10]. Water management is costly, requiring infrastructure, maintenance and often replenishing by vehicle or helicopter. Catchment establishment is institutionalized and continues unabated [2, 3, 11]. Indeed, what began as managing water for quail and mule deer circa 1940 [3, 5] blossomed into ~6,000 managed waters spanning 10 western states at the close of the 20th century [2]. The state of Arizona moved from 750 managed waters for wildlife in 1997 to 3,000 in 2019, with ~1 million annual expenditure [2, 12].

Throughout the United States, the understanding of how water management effects the population demographics of wildlife (e.g., recruitment, survival), or the physiological requirements and water balance of desert ungulates remains founded mostly on correlative studies and anecdotes [3, 5]. Yet globally, water management often fails to demonstrate its value to target game, or generates unintended consequences by increasing predator densities and predation, reducing biodiversity, and promoting range degradation [2, 3, 5, 7, 8]. Hence, Kruger National Park (South Africa) closed many water catchments in the 1990's to promote biodiversity and ecosystem processes [9].

We hypothesized that differences in desert bighorn sheep, mule deer, coyote and mountain lions responses to weather and predator-prey relationships would reveal when and why these species visited water. We used trail cameras for monitoring water catchments primarily established for desert bighorn sheep within the subtropical and hot Chihuahuan, Sonoran and Mojave deserts of the southwest United States, from July 2009 through December 2016. In general, winter precipitation dominates the Mojave Desert, while most precipitation in the Chihuahuan occurs during summer [13]. The Sonoran Desert is hottest, with more

intermediate precipitation during winter and summer [13]. Such differences in weather between deserts may influence when species visit managed waters.

Our regional analyses, spanning these three North American deserts build off prior efforts aimed at understanding when and why desert species visit water sources in the arid environments of North America [6, 14–16]. To our knowledge, our analyses form the most spatially comprehensive and longest investigation of visits to water by desert animals undertaken anywhere.

We identified when desert bighorn sheep were most likely to visit water (i.e. timing [months]), and calculated how many visits occurred during and outside this window for the non-target species. Methodologically, we relied on multi-site, multi-species models within a Bayesian framework to produce biological relationships explaining the visits with 4 weather variables, while incorporating information from visits by sympatric mammals. Our analyses included mule deer, since mountain lions preferentially prey on mule deer throughout the southwest United States, and the presence of mule deer strongly influences mountain lion distributions [17, 18]. Moreover, increased predation by mountain lions on desert bighorn sheep often corresponds with higher abundances of alternative prey, such as mule deer [19–22].

By examining the patterns and causes of species visits to managed waters our applied study provides foundational information for understanding the ecological relationships between weather, predators and prey in the southwestern United States. We offer specific management approaches for improving the efficacy and efficiency of water management, by identifying the potential to target water management to the desired species. Our work exemplifies a responsible, scientific, and ecological approach toward water management aimed at improving wildlife stewardship in the United States and within other arid environments world-wide.

## Materials and methods

We used imagery from remote cameras to examine mammal visits to 105 water locations at nine sites spanning the Chihuahuan, Sonoran and Mojave deserts from July 2009 through December 2016 (Table 1; Fig 1; 3.4 million total images). The Chihuahuan Desert sites included Sevilleta National Wildife Refuge (NWR; latitude 34.30˚N, longitude 106.85˚W; 930.8 km$^2$) and Armendaris Ranch (New Mexico Ranch Properties, Inc.; latitude 33.18˚N, longitude 107.03˚W; 1,500 km$^2$). Here, all water sources established for desert bighorn sheep were monitored (including 8 natural waters), plus additional managed waters within Sevilleta NWR (to improve data acquisition; Table 1). All water sources monitored in the Sonoran and Mojave sites were established for desert bighorn sheep. The three Sonoran sites included Cabeza Prieta NWR (latitude 32.29˚N, longitude 113.36˚W; 4,440 km$^2$; 18 of 21 managed waters monitored) the Barry M. Goldwater Range (East) (BMGR East; latitude 32.60˚N, longitude 113.31˚W; 6,300 km$^2$; 11 of 14 managed waters monitored) and Kofa NWR (latitude 33.26˚N, longitude 114.00˚W; 3,900 km$^2$; 10 of 35 managed waters monitored). The four Mojave Desert locations included the McCullough Mountains (latitude 35.77˚N, longitude 115.12˚W; 470 km$^2$; 6 of 7 managed waters monitored), River Mountains (latitude 36.05˚N, longitude 114.89˚W; 92 km$^2$; all available water sources monitored (6)), Muddy Mountains (latitude 36.34˚, longitude 114.64˚W; 394 km$^2$; 5 of 6 managed waters monitored) and Desert NWR (Sheep Range: latitude 36.46˚N, longitude 115.26˚W; 6,540 km$^2$; 5 of 12 managed waters monitored). When all managed waters were not monitored at a site, then we monitored a random sample given logistics, accessibility, safety considerations and resources to sort the imagery.

The monitored locations occurred in mountainous, rugged terrain favored by desert bighorn sheep [24], except for Sevilleta NWR, where some desert bighorn sheep visited water in a high desert environment outside of rugged terrain. All sites provided water consistently

**Table 1. Metadata of field camera operation.** Columns indicate the number of trail cameras and iButtons (#Cam/iBs), duration of camera operation (with mean and standard deviation [days] across locations), sources used to supplement data (with percent of data supplemented [%]), the number of independent images analyzed by species, including the total number of images acquired (separated by a "/" [desert bighorn sheep (DBS), mule deer (MD), coyote (CY) and mountain lion (MY)]).

| Desert | Site | #Cam/iBs | Start Date | End Date | Mean (SD) | Supplement Source | % | DBS # | MD # | CY # | ML # |
|---|---|---|---|---|---|---|---|---|---|---|---|
| Chihuahuan | Armendaris Ranch | 6/4 | 5/30/2011 | 10/06/2014 | 887(213) | RAWS (Bosque del Apache NWR) | 27 | 828 / 8125 | 531 / 7845 | 22 / 48 | 35 / 56 |
| Chihuahuan | Sevilleta NWR | 38/0 | 7/06/2009 | 7/14/2014 | 1292 (450) | LTER | 0 | 41 / 207 | 3286 / 56698 | 2412 / 9499 | 314 / 487 |
| Mojave | Desert NWR | 5/5 | 8/24/2009 | 2/12/2016 | 1576 (703) | NOAA (Desert NWR); RAWS (Red Rock, NV) | 43 | 256 / 4616 | 279 / 5679 | 339 / 1410 | 81 / 144 |
| Mojave | McCullough Range | 6/5 | 7/12/2011 | 12/19/2016 | 1374 (632) | NOAA (Desert NWR); RAWS (Red Rock, NV) | 36 | 529 / 21133 | 44 / 121 | 324 / 1344 | 67 / 85 |
| Mojave | Muddy Mountains | 5/3 | 1/01/2010 | 12/07/2016 | 1924 (263) | NOAA (Desert NWR); RAWS (Red Rock, NV) | 46 | 664 / 21444 | 0 / 0 | 51 / 71 | 1 / 1 |
| Mojave | River Mountains | 6/2 | 8/18/2011 | 11/17/2016 | 1660 (520) | NOAA (Desert NWR); RAWS (Red Rock, NV) | 37 | 572 / 15878 | 0 / 0 | 667 / 2836 | 0 / 0 |
| Sonoran | BMGR East | 11/6 | 1/3/2011 | 1/26/2015 | 1224 (146) | SDCN | 16 | 474 / 5302 | 156 / 1200 | 499 / 3704 | 48 / 61 |
| Sonoran | Cabeza Prieta NWR | 18/9 | 9/27/2010 | 11/03/2014 | 1103 (264) | SDCN | 5 | 634 / 7321 | 246 / 2869 | 1458 / 10783 | 91 / 132 |
| Sonoran | KOFA NWR | 10/8 | 6/27/2011 | 7/28/2014 | 1106(47) | WRCC | 1 | 309 / 2239 | 296 / 2624 | 303 / 822 | 78 / 185 |

RAWS = https://raws.dri.edu/; LTER = https://lternet.edu; NOAA = https://www.ncdc.noaa.gov/; SDCN = Sonoran Desert Climate Network http://98.191.112.244/; WRCC = Western Regional Climate Center COOP https://wrcc.dri.edu/coopmap/

throughout this study. Every site contained managed waters outside desert bighorn sheep habitats and within their boundaries that were not monitored. No sites contained domestic livestock.

We deployed trail cameras built by Bushnell® (Trophy Cam), Covert (Deuce) and Reconyx (Hyperfire HC550) set 0.9–1.2m above ground, and oriented north whenever possible to minimize sun exposure in the imagery. Although managed waters often differed in size and configuration, we ensured that each cameras field of view encompassed the entire managed water structure. Cameras were motion activated and set with a 10s delay at the Chihuahuan and Sonoran sites, and a 60s delay at the Mohave sites (whereby the camera has a >10s (or 60s) downtime after recording a picture before it can record another). Cameras remained stationary and operated continuously, with rare camera malfunctions quickly addressed. The earliest camera deployment occurred at Sevilleta NWR (07/06/2009) and the latest retrieval in the Mojave Desert (12/19/2016; Table 1). The shortest duration occurred at Kofa NWR (1127 days), with the longest duration at the Muddy Mountains (2532 days). Imagery was sorted according to established protocols [25, 26]. Our study received approval by the United States Air Force, United States Bureau of Land Management, United States Fish and Wildlife Service, United States Geological Survey, United States National Park Service, Nevada Department of Wildlife and New Mexico Ranch Properties Inc., prior to conducting the work. The authors confirm that all methods and experiments were performed in accordance with the relevant guidelines and regulations of these agencies.

## Visit independence

Species visits are typically captured by multiple images. Our analyses required independent visits [6]. The procedure quantifies the amount of time elapsing between sequential images, to

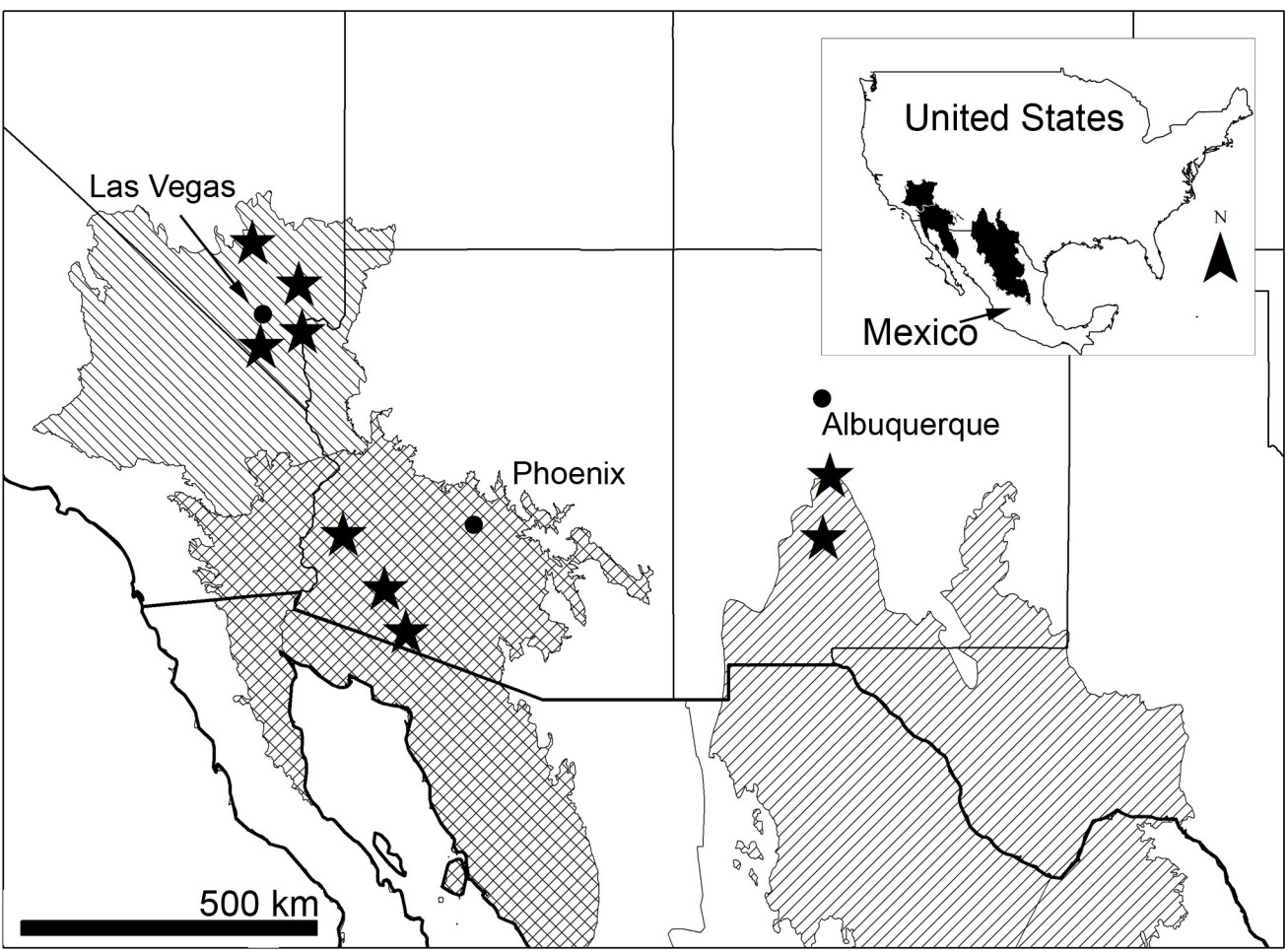

**Fig 1. Location of field sites.** Nine field sites (black stars) within 3 southwestern deserts of the United States (Chihuahuan (far right hatching), Sonoran (crosshatching) and Mojave (far left hatching)), where field cameras recorded imagery describing visits to water for desert bighorn sheep, mule deer, coyotes and mountain lions, 2009–2016 (TNC Ecorgions [23]); Figure generated using ArcGIS Desktop 10.6 software [https://www.esri.com]).

identify a new visit. We considered images collected one hour or more apart at the same location as independent records. This allowed estimating the duration of a visit, and the time between visits for each species (i.e., the time between two independent visits, by the same or different individuals of the same species). The time interval can be any amount provided it exceeds the actual duration that a species visits [6]. For each species, we identified the maximum number of individuals recorded in a single image within each set of independent visits, at each location. We summed these counts within a weekly period, separately for each species and location. This count formed the response variable in analyses (hereafter termed "visit").

## Predictor variables

We analyzed mammal visits to managed waters by relating them with maximum temperature (˚C), relative humidity and precipitation (mm) as explanatory variables. We used maximum temperature data since hotter temperatures raise moisture stress on animals, which may increase their visits [27, 28]. Maximum temperature represents a weekly average of the daily maximum temperature. Relative humidity (RH) describes the amount of moisture content in the air, and animals lose water faster during periods with low RH [28]. Relative humidity is the

weekly average of the daily minimum relative humidity. We incorporated the timing and amount of precipitation in our analyses, because rainfall influences surface water availability, enabling animals to visit water less. Precipitation also increases water content in plants, allowing ungulates to meet their water requirements from forage [28, 29]. Precipitation consisted of two variables: The total amount of precipitation within a week, and the total amount of precipitation during a week and the previous five weeks (a 6-week precipitation total [6]).

Most sites relied on Temperature Logging iButtons (iButton; Table 1) deployed at field camera locations. iButtons recorded temperature and relative humidity at hourly intervals. For each site, locations lacking an iButton used data from the closest neighboring iButton. Precipitation data, and information to supplement gaps in the temperature or relative humidity data were acquired from the closest weather station to the site (Table 1).

## Model structure

We identified when desert bighorn sheep, mule deer, coyote and mountain lions visited water by calculating the mean proportion of visits to water catchments per month, for each site. We grouped results by desert, and calculations only included full years of data. The counts underlying these proportions were corrected by effort, by dividing the total number of species visits by the total number of camera trap days operating during that month.

We then used a hierarchical Bayesian multispecies approach. The model estimates values of species-specific parameters from the spatially replicated observations, by assuming that terms among species and location-level random effects are independent and exchangeable. This approach accounts for the variability in visit number across space (individual camera location) and time. The observed elements consist of species-specific visits, $y$, from each week $j = 1,...,J$ within each set of $i = 1,...,I$ locations for the $k = 1,...,K$ species. The observed data, $y_{ijk}$, was denoted by the matrix of visits for each species as $Y = \{y_{ijk}:i = 1,2,...,I; j = 1,...,J; k = 1,...,K\}$ and regarded as a Poisson outcome $h(y_{ijk}|\lambda_{ijk})$,

$$P\left(y_{ijk} = l\right) = \frac{(\lambda_{ijk})^l}{l!} exp\left(-\lambda_{ijk}\right)$$

where $\lambda_{ijk}$ is the underlying Poisson mean of $y_{ijk}$. Since $\lambda_{ijk}$ varies among locations, the model addresses overdispersion by incorporating a natural hierarchical extension, accounting for extra-Poisson variation. We specify the process model for $\lambda_{ijk}$ to be marginal to a hierarchical element, $\varepsilon_i$ [30]. The dispersion parameter of the hierarchical element is integrated into the likelihood of the Poisson process as a random effect, which accounts for the variation among locations, resulting in a marginally distributed negative binomial outcome $f(N_{ijk}|y_{ijk}\varepsilon_i)$, $\varepsilon_i \sim gamma(\theta,\theta)$ which results in a probability distribution ($P$) for $\lambda_{ijk}$ as

$$P\left(Y_{ijk} = y_{ijk}|\lambda_{ijk}, \theta\right) = \frac{\Gamma(y_{ijk} + \theta)}{\Gamma(y_{ijk} + 1)\Gamma(\theta)} \left(\frac{\lambda_{ijk}}{\lambda_{ijk} + \theta}\right)^{y_{ijk}} \left(\frac{\theta}{\lambda_{ijk} + \theta}\right)^{\theta}$$

The parameter $\theta$ is positive. Large values of $\theta$ generate variability consistent with a Poisson distribution. As $\theta \to \infty$, the distribution converges to a Poisson random variable, where the level of dispersion ($\theta$) is assumed equal among all $I$ camera locations [30]. A generalized linear function is assumed between the mean count $\lambda_{ijk}$ and predictor variables $X_{ij}$:

$$log(\lambda_{ijk}) = \alpha_0 + \sum_{v=1}^{w} \alpha_v X_{v,ij} + \delta(1 + y_{ij-1k}) + \beta(\lambda_{ij-1k})$$

where $X_{ij}$ are predictors $v = 1,2,..,w$ measured at location $i$ within week $j$. The $\alpha's$ are the

intercept and slope parameter estimates. We added a first-order autoregressive process to address the temporal correlation among visits during successive weeks, where $\delta$ and $\beta$ express the dependence of the expected number of new species-specific animal visits, $\lambda_{ijk}$, on the past counts of animals visits with lag-1 dependence [31]. The $\delta$ parameter represents the short-term dependence on the previous time point, whereas the $\beta$ parameter represents the long-term dependence or autocorrelation coefficient on all past values of the observed process.

The log of the number of camera trap days operational during each week served as a linear offset. Covariates were standardized with a mean of zero and standard deviation of one. Only predictor variables with a correlation coefficient ($r$)<0.60 were used in the same model [32]. The models for desert bighorn sheep and mule deer at Sevilleta NWR would not converge with the weekly precipitation and relative humidity variables included, so we excluded those variables from that analysis. In all models, we evaluated the strength of evidence for covariate effects by estimating posterior model probabilities using an inclusion parameter to identify the most probable predictors (i.e.($Pr(X_{ij}>0)$) [31]). We specified the inclusion parameters as latent binary variables (Bernoulli) using a balanced prior probability for each $X_{ij}$, where the prior probability of inclusion for each variable was 0.5. For example, when the posterior probability of inclusion was $X_{ij} = 0$, then the predictor variable had zero effect. If $X_{ij} = 1$, this corresponds to the predictor variable having a high degree of support for being in the "best" model. We considered variables with $Pr(X_{ij}>0.70)$ highly supported and included them in final models [33]. We assessed the discrepancy ($D(N)$) in model adequacy by measuring departures from the observed data $Y_{ijk}$ and assumed model $\lambda_{ijk}$ by calculating a Bayesian $P$ value for each candidate model, by

$$P_D = P[D(\lambda^*, \tau) > D(\lambda, \tau)|\lambda] = \int P[D(\lambda^*, \tau) > D(\lambda, \tau)|\tau]p(\tau|\lambda)d\tau$$

Bayesian P values near 0.5 indicate good model fit. Values close to 0 and 1 suggest lack of fit [34]. We fit models with jagsUI 1.3.1 to access JAGS 3.4.0 using Markov chain Monte Carlo (MCMC) algorithms to generate posterior distributions of the parameters [35, 36]. Models used 4 parallel chains simulated with >150,000 iterations with a burn in of the first 50,000 iterations. We assessed stationarity (chain convergence) with the Gelman-Rubin diagnostic convergence statistics, examination of chain histories, and posterior density plots [37].

We also included visit information from 7 other species to improve the variance calculation on the parameter estimates for the 4 species we focused on herein (coyote, desert bighorn sheep, mountain lion and mule deer). The 7 other species included: bobcat (*Lynx rufus*), black-tailed jackrabbit (*Lepus californicus*), desert cottontail rabbits (*Sylvilagus audubonii*), elk (*Cervis canadensis*), gray fox (*Urocyon cinereoargenteus*), oryx (*Oryx gazella*; nonnative), and pronghorn (*Antilocapra americana*). Data from sites in the Mojave Desert contained the same species enabling data analysis in one model. The Chihuahuan and Sonoran sites included different species so we modeled sites separately.

## Results

In the Chihuahuan Desert, 70% of all desert bighorn sheep visits occurred from April through August (SD 0.09; N = 5 site/years; Fig 2). In the Sonoran Desert, 85% of all desert bighorn sheep visits (SD 0.07; N = 9 site/years) occurred during May through August. In the Mojave Desert, 83% of desert bighorn sheep visits (SD 0.11; N = 21 site/years) occurred from May through September.

These 4–5 month brackets receiving most visits to water for desert bighorn sheep account for the following percentages of mule deer visits, respectively: Chihuahuan 48% (SD 0.11),

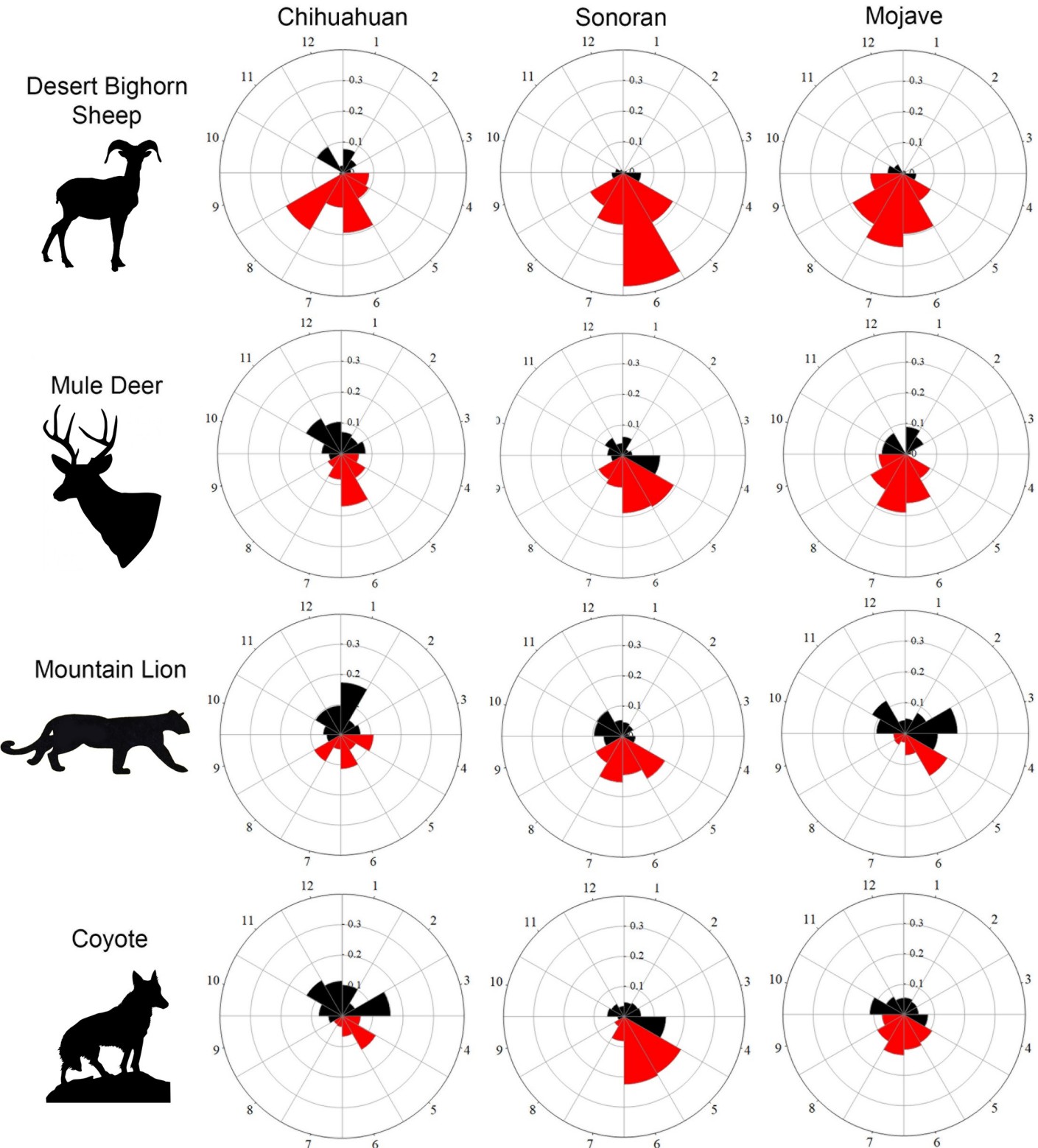

**Fig 2. Polar plots describing the timing of species visits.** Plots report the mean proportion of visits to water catchments per month for 4 species across 3 deserts (data pooled across sites and years within a desert). For all plots, red indicates the 4–5 contiguous months receiving the most visits for desert bighorn sheep (January-December (shown as 1–12)). Black indicates the months receiving the least amount of visits to water catchments by bighorn sheep. Bighorn concentrate visits to a 4–5 month period, while both carnivores and mule deer visit water catchments generally year-round. Species silhouettes from [https://www.needpix.com/] under a Creative Commons Zero License for Public Domain.

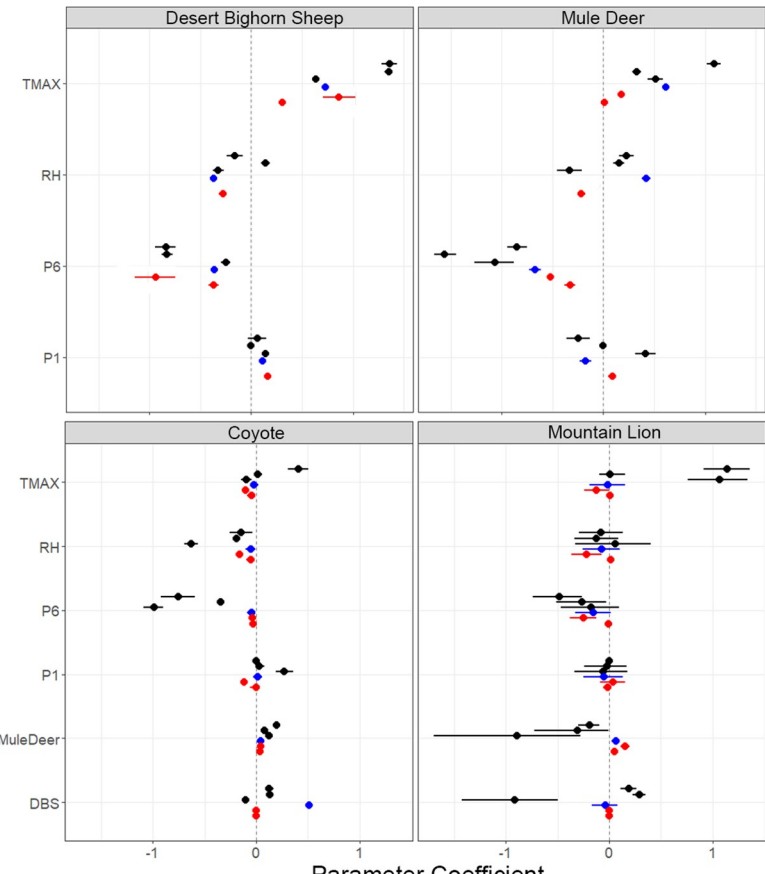

**Fig 3. Slope parameters and 95% credibility intervals.** Results from multi-species, multi-site models predicting species visits to water catchments within three southwestern deserts of the United States (Chihuahan:red; Mojave:blue; Sonoran:black). For each desert, increases in maximum temperature (TMAX; weekly average of daily maximum temperature) and declines in long-term precipitation (P6: weekly sum of precipitation within a week and the prior 5 weeks) raised visits for mule deer and desert bighorn sheep. RH indicates the weekly average of daily minimum relative humidity, and declines in RH associated with more visits to water by coyotes. Visits to water by mountain lions also increased with declines in long-term precipitation. Visits to water by prey were inconsistent predictors of visits to water by mountain lions. P1 represents the weekly sum of precipitation.

Sonoran 57% (SD 0.13), and Mojave 67% (SD 0.28; Fig 2). Percentages of coyote visits included: Chihuahuan 33% (SD 0.19), Sonoran 56% (SD 0.11), and Mojave 52% (SD 0.21; Fig 2). Percentages of visits for mountain lions was Chihuahuan 42% (SD 0.20), Sonoran 54% (SD 0.14), and Mojave 32% (SD 0.27). A species visiting catchments equally throughout a year would record 8.3% of visits per month, or 33% and 42% total visits within 4 and 5 month periods, respectively.

Increased maximum temperature and less long-term (6-week) precipitation always increased visits for desert bighorn sheep and mule deer. Sheep at Cabeza Prieta NWR, Kofa NWR (Sonoran Desert) and Sevilleta NWR (Chihuahuan) displayed the greatest responses to temperature and long-term precipitation (Fig 3).Mule deer displayed the greatest response to temperature at Kofa NWR and the greatest response to long-term precipitation at Cabeza Prieta NWR. Declines in relative humidity and increases in weekly precipitation raised sheep visits to water except at Cabeza Prieta NWR (Fig 3). Mule deer displayed mixed responses to declines in relative humidity and weekly precipitation across deserts and sites (Fig 3).

For mountain lions, less long-term precipitation always associated with increased visits. The response was greatest at Kofa NWR (Fig 3). The other weather variables showed mixed responses. Visits to water by mule deer associated with increased visits to water by mountain lions at the Mojave and Chihuahuan sites and decreased mountain lion visits at the Sonoran sites. Visits to water by desert bighorn sheep associated with increased mountain lion visits to water at Kofa and Cabeza Prieta NWRs, but decreased visits at BMGR East. There existed essentially no relationship between visits to managed waters by desert bighorn sheep and mountain lions at the Mojave and Chihuahuan sites (Fig 3).

Declines in long-term precipitation and relative humidity always associated with increased visits by coyotes, with effects largest at BMGR East (Fig 3). Weekly precipitation and maximum temperature had mixed affects. Increased visits by mule deer to water associated with increases in coyote visits, for every site, with parameters highest at Kofa NWR (Fig 3). An increase in visits to water by desert bighorn sheep associated with increased coyote visits only at the Mojave, plus Cabeza Prieta and Kofa NWRs (Fig 3).

Comparisons between standardized slope parameters revealed further information about the effects of weather on species visits. Mule deer and desert bighorn sheep differed in their responses to long-term precipitation and maximum temperature. Mule deer were typically 1.4 (low) to 10 times (high) more likely to visit water with every unit decline in long-term precipitation than desert bighorn sheep (Sevilleta NWR displayed an opposite pattern). Alternatively, desert bighorn sheep were typically 1.5 to 14 times more likely to visit water with every unit increase in maximum temperature than mule deer (exceptions were at BMGR East and the Mojave sites). Visits to water by mountain lions were influenced by long-term precipitation more than the ungulates. For every unit decline in long-term precipitation, mountain lions were 3 to 37 times more likely (mean 17.3; SD 15.3) to visit water than desert bighorn sheep and 3 to 20 times more likely (mean 7.0; SD 7.3) to visit water than mule deer. The only exception occurred at Armendaris Ranch, where desert bighorn sheep and mule deer were 2 times more likely to visit water with every unit decline in long-term precipitation than mountain lions. Mule deer were typically 2 to 13 times more likely to visit water with every unit decline in precipitation than coyotes, although Armendaris Ranch and Kofa NWR were exceptions. Slope comparisons describing the visits to water by desert bighorn sheep and coyotes varied inconsistently between sites in regards to long-term precipitation.

## Discussion

Improving water management hinges on understanding patterns of species visits, for tailoring water management towards the species justifying catchment establishment [3, 5, 14–16]. Therefore, we present a data-rich study that investigates the response of four predators and prey to water provisioning in three deserts of the southwestern United States. The water management strategy we analyze has been primarily established for increasing populations and distributions of desert bighorn sheep. Our analyses revealed species-specific temporal differences in visits to managed waters. We unraveled why these temporal patterns ensued to advance the understanding of these species' ecologies and interspecific interactions. Our aim is to use these results for advancing water management toward species stewardship.

Lower amounts of long-term precipitation and higher maximum temperatures always increased visits to water by mule deer and desert bighorn sheep (Fig 3). Temperature tended to influence desert bighorn sheep visits more than mule deer. Long-term precipitation influenced visits to water by mule deer more than desert bighorn sheep. Long-term precipitation influenced mountain lion visits to water more than both prey, and coyote visits consistently associated with declines in relative humidity (Fig 3). These differences in species responses to

weather conditions caused mismatches in the timing of visits to water throughout the year. Desert bighorn sheep concentrated their visits to water within 4–5 summer months across all 3 deserts (Fig 2). Mountain lions visited water year-round in the Chihuahuan and Mojave deserts, and generally year-round in the Sonoran (Fig 2). Similar to mountain lions, coyote and mule deer visited water year-round in the Chihuhuan and generally year-round in the other deserts (Fig 2). Irrespective of the differences in the weather across these deserts and sites, visits to water by mule deer and desert bighorn sheep were inconsistent predictors of visits to water by mountain lions. Mule deer visits always associated with visits to water by coyotes.

Long-term precipitation influenced visits to water by mountain lions more than the prey at all sites except Armendaris Ranch. We suspect this exception occurred because this ranch abuts a dammed reservoir (Elephant Butte) of the Rio Grande. Distances between the water catchments designed for sheep and this reservoir are short (~3.5 km). This situation may allow mountain lions to visit the reservoir, thereby dampening the effects of precipitation on their visits to water catchments.

For the predator models, a positive association with prey (i.e., positive slope parameter) can only occur when the visits to water for a given predator and prey happen at the same location (water catchment) within the same weekly periods, within a given site. Negative associations between visits of predators and prey occur under two different situations. First, if the visit data collected for a predator and prey at the same locations within a given site occur at different weekly periods. Second, if a predator and prey visit water at different locations within a given site, irrespective of whether the visits occur during the same or different weekly periods. For example, at Cabeza Prieta NWR and BMGR East (Sonoran), 94% and 98% of all mule deer visits occurred at catchments with no mountain lion visits, respectively. Meanwhile, 84% and 82% of all mountain lion visits occurred at catchments with no mule deer visits. Similarly, at BMGR East, 1% of all desert bighorn sheep visits were recorded at the sites where 82% of all mountain lion visits occurred. These situations produced negative relationships for visits to water for the predator and prey at these sites, and their causes may be behavioral, physiological or habitat related [38]. Examination of these topics was beyond the scope of our investigation.

A hypothesis leading to indirect management costs of year-round water provisioning includes changes in the range expansion and habitat use of predators. Managed waters allow populations of desert bighorn sheep to inhabit areas they previously had not [3, 39, 40]. Indeed, this outcome forms justification of managing waters for desert bighorn sheep. It follows that managed waters could enable mountain lions to inhabit locations they previously had not, provided a sufficient prey base exists (i.e., as discussed in [41]). Such range expansion of non-target species forms a major unintended consequence of water management [1, 7–9]. If water was provided seasonally when desert bighorn sheep visit it, and not year-round, perhaps predators may temporarily vacate areas, thereby reducing predation on desert bighorn sheep.

Outside of North America, ramifications from managed waters vary. Throughout Africa and Australia, artificial waterholes form centers for predation [8, 42]. Water provision drives lion (*Panthera leo*) movements, habitat selection and hunting behavior [1, 42–44]. Water management at Kruger National Park encouraged zebra (*Equus quagga*) and blue wildebeest (*Connochaetes taurinus*) to inhabit new areas [1]. With water more available, African lions (*Panthera leo*) immigrated. Their predation contributed to a 10-fold decline in roan antelope (*Hippotragus equinus*) that are less reliant on surface water, and inhabited the region prior to water management [1]. Introduced feral cats (*Felis catus*), foxes (*Vulpes vulpes*) and dingos (*Canis lupus*) commonly occur at managed waters causing declines in Australian fauna [8, 45]. Dingos frequent watering points for hunting opportunities, forming the main regulators of kangaroo populations in eastern Australia [46]. Similarly, invasive cane toads (*Bufo marinus*)

use year-round managed waters as dry season refuges, facilitating their spread in Australia [47]. Excluding toads' access to managed waters during the dry seasons removes this resource subsidy, thereby containing its range expansion [47].

We recommend that biologists identify specific objectives for water management. If the focus is a target species (or guild), biologists can use experimentation and monitoring to ensure water management maximizes its assistance to the species. As such, our work models a scientific approach for revealing species associations with managed waters throughout arid environments. We inform water management by identifying the biotic and abiotic factors influencing visits to water by predators and prey. Our results indicate that year-round water provision for desert bighorn sheep appears unnecessary, given their concentration of visits centered on a 4–5 month period. Instead, year-round water management corresponds with the pattern of visits to water for mountain lions, mule deer and coyotes (Figs 2 and 3).

Wildlife managers and ecologists hold responsibility to ensure their management actions improve wildlife stewardship without generating detrimental consequences. Ecologically, we know that management actions can improve habitat conditions for some species while decreasing habitat conditions for others [48]. Therefore, further scientific approaches to water management should examine related testable hypotheses, like if restricting the period of water availability to match visits by desert bighorn sheep (i.e., target species) reduces predator densities and predation risk on desert bighorn sheep, while ensuring water availability during arid months when desert bighorn sheep visit water most. Answering these questions will continue advancing the stewardship of desert mammals in the Southwestern United States while informing water management strategies for species inhabiting arid regions elsewhere.

## Supporting information

**S1 File. Model parameter estimates.** This file contains model parameter estimates (MeanB), with lower and upper 95% credibility intervals (LWRCR2.5, UPRCR97.5) for 4 mammals inhabiting 3 deserts within the southwestern United States.
(XLSX)

**S2 File. Data used in manuscript.** This file contains all relevant data to replicate study findings reported in the article.
(XLSX)

## Acknowledgments

B. Derango, E.D. Edwards, R. Hastings, H. Prude and H. Shaw were extremely helpful and generous with ideas, camera establishment, imagery identification and site access. K. Longshore provided considerable assistance with project conceptualization, data collection, curation and writing. The findings and conclusions in this article are those of the author(s) and do not necessarily represent the views of the U.S. Fish and Wildlife Service. The use of trade, firm, or product names is for descriptive purposes only and does not imply endorsement by the U.S. Government.

## Author Contributions

**Conceptualization:** Grant M. Harris, David R. Stewart, David Brown, Lacrecia Johnson, Jim Sanderson, Aaron Alvidrez, Tom Waddell, Ron Thompson.

**Data curation:** Grant M. Harris, Lacrecia Johnson, Jim Sanderson, Aaron Alvidrez.

**Formal analysis:** Grant M. Harris, David R. Stewart, David Brown, Lacrecia Johnson, Jim Sanderson.

**Investigation:** Grant M. Harris, Lacrecia Johnson, Jim Sanderson, Aaron Alvidrez, Tom Waddell, Ron Thompson.

**Methodology:** Grant M. Harris, David R. Stewart, Lacrecia Johnson, Jim Sanderson, Aaron Alvidrez, Tom Waddell, Ron Thompson.

**Project administration:** Grant M. Harris, Lacrecia Johnson, Jim Sanderson.

**Supervision:** Grant M. Harris, Lacrecia Johnson, Jim Sanderson.

**Validation:** Grant M. Harris, Jim Sanderson.

**Visualization:** Grant M. Harris.

**Writing – original draft:** Grant M. Harris, David R. Stewart, David Brown, Lacrecia Johnson, Jim Sanderson, Aaron Alvidrez, Tom Waddell, Ron Thompson.

**Writing – review & editing:** Grant M. Harris, David R. Stewart, David Brown, Lacrecia Johnson, Jim Sanderson, Aaron Alvidrez, Tom Waddell, Ron Thompson.

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
