## [Decision Letter · Decision Letter 0]

1 Sep 2020

PONE-D-20-24037

Year-round water management for desert bighorn sheep corresponds with visits by predators not bighorn sheep

PLOS ONE

Dear Dr. Harris,

Thank you for submitting your manuscript to PLOS ONE. After careful consideration, we feel that it has merit but does not fully meet PLOS ONE’s publication criteria as it currently stands. Therefore, we invite you to submit a revised version of the manuscript that addresses the points raised during the review process.

Please, check the comments from reviewers, answer their questions and requests. Specific comments from reviewer #2 are attached to this email.

We look forward to receiving your revised manuscript.

Kind regards,

Paulo Corti, Ph.D.

Academic Editor

PLOS ONE

4. We note that Figure 1 in your submission contain map images which may be copyrighted. All PLOS content is published under the Creative Commons Attribution License (CC BY 4.0), which means that the manuscript, images, and Supporting Information files will be freely available online, and any third party is permitted to access, download, copy, distribute, and use these materials in any way, even commercially, with proper attribution. For these reasons, we cannot publish previously copyrighted maps or satellite images created using proprietary data, such as Google software (Google Maps, Street View, and Earth). For more information, see our copyright guidelines: http://journals.plos.org/plosone/s/licenses-and-copyright.

4.1.    You may seek permission from the original copyright holder of Figure 1 to publish the content specifically under the CC BY 4.0 license. 

4.2.    If you are unable to obtain permission from the original copyright holder to publish these figures under the CC BY 4.0 license or if the copyright holder’s requirements are incompatible with the CC BY 4.0 license, please either i) remove the figure or ii) supply a replacement figure that complies with the CC BY 4.0 license. Please check copyright information on all replacement figures and update the figure caption with source information. If applicable, please specify in the figure caption text when a figure is similar but not identical to the original image and is therefore for illustrative purposes only.

Reviewers' comments:

Reviewer's Responses to Questions

**Comments to the Author**

1. Is the manuscript technically sound, and do the data support the conclusions?

Reviewer #1: Yes

Reviewer #2: Yes

2. Has the statistical analysis been performed appropriately and rigorously? 

Reviewer #1: Yes

Reviewer #2: Yes

3. Have the authors made all data underlying the findings in their manuscript fully available?

Reviewer #1: Yes

Reviewer #2: Yes

4. Is the manuscript presented in an intelligible fashion and written in standard English?

Reviewer #1: Yes

Reviewer #2: Yes

5. Review Comments to the Author

Reviewer #1: I enjoyed reading this paper, based on a formidable long-term and multi-site data base. It is well-written and provides relevant data to an important issue, applicable outside the study system. I only have a few minor comments and requests for clarifications.

Specific comments:

Grammatical mishaps and word processing accidents: L. 24 (canadensis lower case); L. 52: prey upon (predate means dated earlier); 81: anecdotes; 81: its value; 99: desert.

L. 41-43: Here it is not too clear what those opportunities are, can this be made more explicit?

Table 1: I am confused by this legend. I do not understand what 'supplemented data' are. Is it referring to water supplementation? Also, I do not understand why the sample size of images at a given location differs by species. Were not all images analyzed for all species?

L. 167-169: So a 'visit' is the cumulative number of maximum counts from all 'independent visits' summed over a week?

L. 244-246: I do not understand this sentence. Does it refer to the 4 study species, or to the other species listed here?

L. 363-365: This sentence is confusing. Only 1% of sheep were seen at sites where 82% of cougar pictures were taken?

L. 368-369: OK, but is there any evidence for or against the idea that predation risk increases at watering sites? If so, it would be good to mention it here.

L. 387: It may be worth pointing out the role of managed water in the expansion of cane toads in Northern Australia. See Florance et al. https://www.ncbi.nlm.nih.gov/pmc/articles/PMC3151714/

Excluding access to invasion hubs can contain the spread of an invasive vertebrate.

Marco Festa-Bianchet

Reviewer #2: I suggest that you review the article of Escobar Flores et al. 2019 : Waterhole detection using a vegetation index in desert bighorn sheep (Ovis canadensis cremnobates) hábitat. The support information of this article mentions a coincidence of the visits of waterholes by bighorn and predators.

6. PLOS authors have the option to publish the peer review history of their article (what does this mean?). If published, this will include your full peer review and any attached files.

Reviewer #1: **Yes: **Marco Festa-Bianchet

Reviewer #2: **Yes: **Sarahi Sandoval

---

## [Author Response · Author response to Decision Letter 0]

5 Oct 2020

Dear PLOS ONE and Dr. Corti,

The "Response to Reviewers" is included as an attached file. I pasted it below, but it is difficult to read in the viewer. The text below and in the attachment are identical.

Thank you -- Grant Harris

Response to Reviewers

Dear PLOS ONE and Dr. Corti,

We appreciate the reviews of our manuscript “Year-round water management for desert bighorn sheep corresponds with visits by predators not bighorn sheep” (PONE-D-20-24037). All of the review comments and suggestions were constructive (black text), and we responded to them below (dark blue text). We took this review seriously and modified our manuscript accordingly. We believe this peer review process has strengthened the influence and utility of our work. 

This document begins with attending to journal specific requirements and then we address the Reviewer comments.

A) Please ensure that your manuscript meets PLOS ONE's style requirements, including those for file naming. 

We ensured that our manuscript meets all PLOS ONE's style requirements, including those for file naming. We apologize in advance if any formatting errors remain.

We recognize that PLOS ONE does not perform copy editing, so we reviewed our paper carefully. In doing so we improved some descriptions in the Method section, enhanced explanation for one of the equations, and replaced another equation with a text description. These minor enhancements improved the clarity of our work, and do not change our approach, methods or results. We also ensured consistent terminology within the MS.

B) In your Methods section, please provide additional information regarding the permits you obtained for the work. Please ensure you have included the full name of the authority that approved the field site access and, if no permits were required, a brief statement explaining why.

We included this information in Methods, on Lines 167 – 172, namely: “Our study received approval by the United States Air Force, United States Bureau of Land Management, United States Fish and Wildlife Service, United States Geological Survey, United States National Park Service, Nevada Department of Wildlife and New Mexico Ranch Properties Inc., prior to conducting the work. The authors confirm that all methods and experiments were performed in accordance with the relevant guidelines and regulations of these agencies.”

C) We note that you have stated that you will provide repository information for your data at acceptance. Should your manuscript be accepted for publication, we will hold it until you provide the relevant accession numbers or DOIs necessary to access your data. If you wish to make changes to your Data Availability statement, please describe these changes in your cover letter and we will update your Data Availability statement to reflect the information you provide.

We decided to change our Data Availability statement, by including these data as a file in supporting information. This file contains all relevant data to replicate study findings reported in the article.

D) We note that Figure 1 in your submission contain map images which may be copyrighted. All PLOS content is published under the Creative Commons Attribution License (CC BY 4.0), which means that the manuscript, images, and Supporting Information files will be freely available online, and any third party is permitted to access, download, copy, distribute, and use these materials in any way, even commercially, with proper attribution. For these reasons, we cannot publish previously copyrighted maps or satellite images created using proprietary data, such as Google software (Google Maps, Street View, and Earth). 

Regarding Figure 1: We generated this figure using ArcGIS Desktop 10.6 software (https://www.esri.com). We edited the manuscript by stating the software used to generate the figure, and citing the source of ecosystem boundary information within the legend. 

After submitting the revision, the PLOS internal check by Ms. Fodor asked about the copyright status for the animal silhouettes in Figure 2. These images are royalty free, which I misconstrued to mean copyright free. Therefore, we found animal silhouettes from a location that has images freely available under the Creative Commons Zero License for public domain (https://www.needpix.com/). We replaced the silhouettes in Figure 2 and submitted that revised figure. Data within the figure, and the plot representations, remain unchanged.

Reviewer #1:

1. I enjoyed reading this paper, based on a formidable long-term and multi-site data base. It is well-written and provides relevant data to an important issue, applicable outside the study system. I only have a few minor comments and requests for clarifications.

Thank you for your review. We feel honored that you found our study interesting and contributory.

Specific comments:

2. Grammatical mishaps and word processing accidents: L. 24 (canadensis lower case); L. 52: prey upon (predate means dated earlier); 81: anecdotes; 81: its value; 99: desert.

Thank you for catching these. We fixed all of them.

3. L. 41-43: Here it is not too clear what those opportunities are, can this be made more explicit? Potential, possibilities, changes in techniques.

We revised the sentence and it now reads on Lines 41 – 43: “Our results suggest improvements to water management by aligning water provision with the patterns and ecological explanations of desert bighorn sheep visits.”

4. Table 1: I am confused by this legend. I do not understand what 'supplemented data' are. Is it referring to water supplementation? Also, I do not understand why the sample size of images at a given location differs by species. Were not all images analyzed for all species?

On lines 200-202 we defined supplemented data. We clarified this sentence, however, in the revision (lines 200-202). We also changed the column heading in Table 1 from "Supplement" to "Supplement Source".

Correct, we analyzed all images to identify them by species. Each camera location, however, imaged different numbers of individual animals by species. For example, camera "A" may have imaged 100 mule deer and 50 desert bighorn sheep, while camera "B" imaged 25 mule deer and 200 desert bighorn sheep. The differences and variability in counts across locations are accounted for with the multi-site, multi-species Bayesian hierarchical models we used for analyses.

5. L. 167-169: So a 'visit' is the cumulative number of maximum counts from all 'independent visits' summed over a week?

Correct. We further clarified on Lines 180 – 183: “For each species, we identified the maximum number of individuals recorded in a single image within each set of independent visits, at each location. We summed these counts within a weekly period, separately for each species and location. This count formed the response variable in analyses (hereafter termed “visit”).”

6. L. 244-246: I do not understand this sentence. Does it refer to the 4 study species, or to the other species listed here?

We revised this section to make this information clear. This section now reads (Lines 261 – 263): “We also included visit information from 7 other species to improve the variance calculation on the parameter estimates for the 4 species we focused on herein (coyote, desert bighorn sheep, mountain lion and mule deer) The 7 other species included:…”

7. L. 363-365: This sentence is confusing. Only 1% of sheep were seen at sites where 82% of cougar pictures were taken?

Good clarification. We reworded this sentence using the suggested text. Lines 382 – 384 now read: “Similarly, at BMGR East, 1% of all desert bighorn sheep visits were recorded at the sites where 82% of all mountain lion visits occurred.” 

8. L. 368-369: OK, but is there any evidence for or against the idea that predation risk increases at watering sites? If so, it would be good to mention it here.

We have been unable to locate any published evidence for or against the idea that predation risk by mountain lions increases at managed waters. We are, however, working on such a study. We hope to have the work completed and submitted in another 6 months or so. 

9. L. 387: It may be worth pointing out the role of managed water in the expansion of cane toads in Northern Australia. See Florance et al. https://www.ncbi.nlm.nih.gov/pmc/articles/PMC3151714/

Excluding access to invasion hubs can contain the spread of an invasive vertebrate.

Thank you for bringing this interesting study to our attention. We included this work on lines 407 – 410.

Marco Festa-Bianchet

Reviewer #2: 

10. I suggest that you review the article of Escobar Flores et al. 2019 : Waterhole detection using a vegetation index in desert bighorn sheep (Ovis canadensis cremnobates) hábitat. The support information of this article mentions a coincidence of the visits of waterholes by bighorn and predators.

Escobar Flores et al. 2019 present an important study focusing on the detection of waterholes using NDVI and NDWI indices. Although the supplemental material for this study reports images of predators and prey at water sites, the publication does not contain any analyses of those data. Therefore we did not refer to this publication within this MS.

11. Abstract Line 23. Add lowercase to the beginning of the scientific name

Thank you. It’s been fixed.

12. Introduction Line 53. I suggest that you review the article of Escobar Flores et al. 2019 : Waterhole detection using a vegetation index in desert bighorn sheep (Ovis canadensis cremnobates) hábitat. The support information of this article mentions a coincidence of the visits of waterholes by bighorn and predators.

Please refer to the response for comment #10.

13. Materials and methods Line 116. Please datial the characteristics of the camera for example: trigger speed, setting that capture per trigger…etc….Examples of this can be found in the following articles: Perry TW, Newman T, Thibault KM. Evaluation of methods used to estimate size of a population of desert bighorn sheep (Ovis canadensis mexicana) in New Mexico. Southw Nat. 2010;55: 517–524. Whiting, J. C., Bowyer, R. T., Flinders, J. T., Bleich, V. C., & Kie, J. G. (2010). Sexual segregation and use of water by bighorn sheep: implications for conservation. Animal Conservation, 13(6), 541-548. Escobar-Flores, J. G., Sandoval, S., Valdez, R., Shahriary, E., Torres, J., Alvarez-Cardenas, S., & Gallina-Tessaro, P. (2019). Waterhole detection using a vegetation index in desert bighorn sheep (Ovis canadensis cremnobates) habitat. PloS one, 14(1), e0211202.

Good idea. We enhanced this section in the Methods by including this information on lines 156- 167.

14. Line 174. Please mention with which instrument the temperature, relative humidity and precipitation were measure.

The sources for these data were identified on Lines 198 - 202. We revised this text to help ensure clarity. Please refer to lines 198 - 202.

---

## [Editor Report · Decision Letter 1]

9 Oct 2020

Year-round water management for desert bighorn sheep corresponds with visits by predators not bighorn sheep

PONE-D-20-24037R1

Dear Dr. Harris,

We’re pleased to inform you that your manuscript has been judged scientifically suitable for publication and will be formally accepted for publication once it meets all outstanding technical requirements.

Kind regards,

Paulo Corti, Ph.D.

Academic Editor

PLOS ONE
---

## [Editor Report · Acceptance letter]

13 Nov 2020

PONE-D-20-24037R1 

Year-round water management for desert bighorn sheep corresponds with visits by predators not bighorn sheep 

Dear Dr. Harris:

I'm pleased to inform you that your manuscript has been deemed suitable for publication in PLOS ONE. Congratulations! Your manuscript is now with our production department. 

Kind regards, 

on behalf of

Dr. Paulo Corti 

Academic Editor

PLOS ONE